# Analysis of the Frequency Shift versus Force Gradient of a Dynamic AFM Quartz Tuning Fork Subject to Lennard-Jones Potential Force

**DOI:** 10.3390/s19081948

**Published:** 2019-04-25

**Authors:** Chia-Ou Chang, Wen-Tien Chang-Chien, Jia-Po Song, Chuang Zhou, Bo-Shiun Huang

**Affiliations:** 1College of Mechanical Engineering, Guangxi University, Nanning 530004, China; songarpore@gmail.com (J.-P.S.); zhouchuangzw@163.com (C.Z.); 2Institute of Applied Mechanics, National Taiwan University, Taipei 106, Taiwan; d98543013@ntu.edu.tw; 3Department of Information Technology and Management, Fooyin University, Tai-Liao, Kaohsiung 831, Taiwan; sc102@fy.edu.tw

**Keywords:** quartz tuning fork, atomic force microscopy, frequency shift, force gradient, Hamilton’s principle

## Abstract

A self-sensing and self-actuating quartz tuning fork (QTF) can be used to obtain its frequency shift as function of the tip-sample distance. Once the function of the frequency shift versus force gradient is acquired, the combination of these two functions results in the relationship between the force gradient and the tip-sample distance. Integrating the force gradient once and twice elucidates the values of the interaction force and the interatomic potential, respectively. However, getting the frequency shift as a function of the force gradient requires a physical model which can describe the equations of motion properly. Most papers have adopted the single harmonic oscillator model, but encountered the problem of determining the spring constant. Their methods of finding the spring constant are very controversial in the research community and full of discrepancies. By circumventing the determination of the spring constant, we propose a method which models the prongs and proof mass as elastic bodies. Through the use of Hamilton’s principle, we can obtain the equations of motion of the QTF, which is subject to Lennard-Jones potential force. Solving these equations of motion analytically, we get the relationship between the frequency shift and force gradient.

## 1. Introduction 

Quartz tuning forks (QTF) are widely used as a scanning probe for atomic force microscopy (AFM). Due to the inherent precise oscillation frequency of single-crystal quartz, the frequency shift is measured as a signal for the atomic and subatomic resolution images of surface of metals, dielectrics, and semiconductors [1,2]. Pawlak et al. [3] used QTF-based non-contact AFM (nc-AFM) to observe the island growth process and the hexagonal structure of fullerene assembly on a KBr(001). They found that Cu(111) substrate has a stronger molecule-substrate interaction than KBr(001), so that Cu(111) substrate can hold the fullerene and not be disrupted by the nc-AFM when its tip approaches the short-range repulsive force regime. The first time that individual atoms within the aromatic molecules were resolved was done by means of nc-AFM [4]. Both techniques of nc-AFM and STM (scanning tunneling microscope) can be used complementarily to study atomic bonding within and between molecules, molecular orbitals, molecular conformational changes, and chemical reactions at the single-molecule level [4,5]. Polesel-Marris et al. [5] used nc-AFM to observe the unfolding signatures during protein stretching and they also functionalized the tip, so as to graft proteins onto the substrate. The force-distance curve of AFM can be used to detect the stiffness change of diseased cells compared with the surrounding cell wall [6]. 

Certain researchers have enhanced the functions of AFM by modifying the structure of a QTF probe [7,8,9,10,11]. Bayat [7] coupled a cantilever probe called the Akiyama probe to a QTF. The Akiyama probe oscillating at its second eigenmode can be used to measure soft samples in tapping mode, while the QTF oscillating at its first eigenmode is used to operate in either tapping mode or non-contact mode. Hussain [9] modified the structure of the tip and its orientation, such that the QTF can perform sidewall roughness measurements. 

Although the output of nc-AFM can provide the frequency shift as a function of the distance between the tip and sample [12,13,14,15,16], the tip-sample interaction force cannot be obtained directly from this kind of experiment. First, we need the equations of motion, which can describe exactly the motion behavior of the system composed of a QTF and the tip-sample interaction force. Tip-sample interaction force is equal to the negative gradient of the tip-sample potential and the gradient of the force (or force gradient) is the spring constant of the physically-modeled spring connecting the tip and sample. By solving the equations of motion, the frequency shift as a function of the force gradient can be obtained. The combination of these two functions yields the force gradient as a function of the tip-sample distance. Then, the integration of the force gradient provides the tip-sample interaction force and the second integration provides the interaction potential. 

References [13,14,15] have modeled a QTF as a simple one-dimensional harmonic oscillator and they use their own method for estimating the spring constant of this oscillator. Castellanos-Gomez [17] modeled the QTF as two collinear coupling oscillators and he evaluated the spring constant of a single prong, as well as the spring constant, of the coupling spring connecting the two prongs by theoretical and experimental analysis. Falter et al. [18] calibrated the beam formula for determining the static spring constant of a qPlus sensor by shifting the origin of the prong to the zero stress zone inside the basis and including the torsion effect. Melcher et al. [19] analyzed a qPlus QTF sensor, in which the tip is of finite length and has axial offset. They improved the accuracy of the spring constant estimation by correct derivation of the angular displacement of the finite tip during the deformation of the slender prong, and also by considering the effect of the peaked ridges appearing in the side wall of the prongs. In this paper, we model the two prongs and the proof mass as continuum bodies instead of the spring-mass oscillator model, and establish the equations of motion through the use of Hamilton’s principle, including the Lennard-Jones potential force between the tip and sample. Although our equations of motion are more complicated than those of harmonic oscillators, there is no need for us to determine the equivalent spring constant. 

The free prong, the prong loaded with interaction force, and the proof mass are taken apart. With the interface surfaces acted on by shear forces and moments of opposite direction, their own equations of motion are set up independently, and their general solutions are solved analytically. Finally, putting all these three sets of general solutions together with the boundary and interface conditions imposed results in the characteristic equation, which can be solved to get the natural frequency of the in-plane anti-phase resonant mode of the QTF. Based on this, the frequency shift, which is the difference between the loaded QTF and free QTF, can be obtained. 

## 2. The Equations of Motion and the General Solution of the Prong Loaded with Tip-Sample Interaction Force

### 2.1. The Linearization of the Lennard-Jones Potential Force

The potential function of Lennard-Jones type is considered here as [20,21,22]
(1)$$W(z)=-\frac{2}{3}\pi^{2}\epsilon \rho_{1}\rho_{2}\sigma^{5}R[\frac{\sigma }{s}-\frac{\sigma^{7}}{210s^{7}}]\;$$
where ϵ is the dielectric constant, σ is the radius of the sample molecule, R is the radius of the atom of the tip, $\rho_{1}\;\mathrm{and}\;\rho_{2}$ are the molecule density of the tip and sample, respectively, and s is the distance between the tip and sample. The interaction force is the gradient of the potential
(2)$$F=-\frac{\partial W}{\partial s}=-\frac{A_{1}R}{180s^{8}}+\frac{A_{2}R}{6s^{2}}$$

Referring to Figure 1, the parameter z is the initial height between the tip and sample, with the interaction force not yet considered. $w^{*}$ is the displacement of the tip in static equilibrium, $\bar{w}\left(x,t\right)$ is the displacement of vibration around the equilibrium state. So the actual distance between the tip and sample in the deformed state of the QTF is (z−w(x,t)) where
(3)$$w\left(x,t\right)=w^{*}\left(x\right)+\bar{w}\left(x,t\right)$$
is the total displacement of the tip. Therefore, Equation (2) becomes
(4)$$F=-\frac{A_{1}R}{180{\left(z-w\right)}^{8}}+\frac{A_{2}R}{6{\left(z-w\right)}^{2}}$$
where $A_{1}=\pi^{2}\rho_{1}\rho_{2}c_{1}$ and $A_{2}=\pi^{2}\rho_{1}\rho_{2}c_{2}$ are called Hamaker constants. In the practical application of nc-AFM, the amplitude $\bar{w}$ is kept small, thus, w≪z, so we can rewrite (4) as
(5)$$F(w)=-\frac{1}{180}A_{1}Rz^{-8}{\left(1-\frac{w}{z}\right)}^{-8}+\frac{1}{6}A_{2}Rz^{-2}{\left(1-\frac{w}{z}\right)}^{-2}$$

Taking a binomial expansion of those terms ${\left(1-w/z\right)}^{-8}$ and ${\left(1-w/z\right)}^{-2}$, Equation (5) reduces to
(6)$$F\left(w;z\right)=k\left(z\right)w\left(x,t;z\right)+F_{0}\left(z\right)$$
where
(7)$$k=(-\frac{2}{45}A_{1}Rz^{-9}+\frac{1}{3}A_{2}Rz^{-3}),\;F_{0}=-\frac{1}{180}A_{1}Rz^{-8}+\frac{1}{6}A_{2}Rz^{-2}$$

Notice that given a value of z in (7), k, and $F_{0}$ becomes constant, and Equation (6) reveals that k plays the role of the spring constant of a linear spring and the Lennard-Jones force can be represented by the linear spring plus a constant force, as shown in Figure 2. 

Taking a derivative of Equation (6) with respect to displacement w gives
(8)$$k=\frac{\partial F}{\partial w}$$

Since s=z−w, for a given value of the initial height z
ds=−dw, then Equation (8) becomes
(9)$$k=\frac{\partial F}{\partial w}=\frac{\partial F}{\partial s}\frac{ds}{dw}=-\frac{\partial F}{\partial s}.$$

Thus the spring constant has the physical meaning of negative force gradient. 

### 2.2. The Deformation and Equations of Motion 

The QTF is made of a Z−cut 2° ingle-crystal wafer. The second-order stress $\sigma_{ij}$ and strain tensors $\varepsilon_{ij}$ are expressed in the vector form as $\left(\sigma_{1},\sigma_{2},\sigma_{3},\sigma_{4},\sigma_{5},\sigma_{6}\right)=$$\left(\sigma_{11},\sigma_{22},\sigma_{33},\sigma_{23},\sigma_{13},\sigma_{12}\right)$, $\left(S_{1},S_{2},S_{3},S_{4},S_{5},S_{6}\right)=\;\left(\varepsilon_{11},\varepsilon_{22},\varepsilon_{33},\varepsilon_{23},\varepsilon_{13},\varepsilon_{12}\right)$. The constitutive law for stress and strain is σ=CS, the stiffness matrix C of $Z-cut{\;2}^{\circ }$ quartz is [23]
(10)$$c^{E}=\left[\begin{array}{cccccc} 86.74 & 6.99 & 11.91 & -17.91 & 0 & 0\\ 6.99 & 86.74 & 11.91 & 17.91 & 0 & 0\\ 11.91 & 11.91 & 107.2 & 0 & 0 & 0\\ -17.91 & 17.91 & 0 & 57.94 & 0 & 0\\ 0 & 0 & 0 & 0 & 57.94 & -17.91\\ 0 & 0 & 0 & 0 & -17.91 & 39.88 \end{array}\right]\times 10^{9}\;N/m^{2}$$

Since the lower prong loaded with a tip-sample potential force is slender, its deformation can be modeled by the well-known Euler beam theory, that is, the cross section perpendicular to the neutral axis remains perpendicular during deformation. Its free body diagram is shown in Figure 3. Here, the conjunctive surface of the proof mass and the prong is assumed to be rotatable and has no $x_{1}$ component of displacement vector. The external shear force $Q_{ext}$ and moment $M_{ext}$, acting at the left boundary, are exerted by the proof mass. The parameters $L_{b},\;b_{b},\;\mathrm{and}\;h$ denote the length, width, and height of the rectangular cross-section of the prong. For the QTF of $\left(ZYw\right)+2^{\circ }$ layout [24], the $x_{2}-\mathrm{axis}$ is designated to be along the longitudinal direction of the prong and the $x_{3}-\mathrm{axis}$ is perpendicular to the QTF surface. 

Let $w\left(x_{2},t\right)$ denote the transverse displacement of the central line of the prong and the displacement field $\left(u_{1},u_{2},u_{3}\right)$ of any point $\left(x_{1},x_{2},x_{3}\right)$ of the prong based on the Euler beam theory is
(11)$$u_{1}=w\left(x_{2},t\right),\;u_{2}=-x_{1}w_{,2}\left(x_{2},t\right),\;u_{3}=0$$
where $w_{,2}=\partial w/\partial x_{2}$. Then the strain fields are
(12)$$S_{1}=0,\;S_{2}=-x_{1}w_{,22},\;S_{3}=S_{4}=S_{5}=S_{6}=0$$

The velocity of point $\left(x_{1},x_{2},x_{3}\right)$is
(13)$$v={\dot{u}}_{1}i+{\dot{u}}_{2}j+{\dot{u}}_{3}k=\frac{\partial w\left(x_{2},t\right)}{\partial t}i-x_{1}\frac{\partial^{2}w\left(x_{2},t\right)}{\partial x_{2}\partial t}j$$

The Hamilton’ principle for an elastic body is
(14)$$\int_{t_{i}}^{t_{f}}\left[\delta T-\delta U+\delta W^{\left(m\right)}\right]dt=0$$
where T is the kinetic energy, U the elastic energy, and $W^{\left(m\right)}$ the work done by the applied mechanical force. At the interface, the moment $M_{ext}$ has the contribution $M_{ext}{\delta \left(w_{,2}\right)|}_{x_{2}=0}$ to $\delta M^{\left(m\right)}$ and no contribution for the shear force $Q_{ext}$. Because there is no $x_{1}-\mathrm{axis}$ component of displacement at $x_{2}=0$, the spring force and constant force $F_{0}$ have done virtual work, due to the virtual displacement ${\delta w|}_{x_{2}=L_{b}}$. So
(15)$$\delta W^{\left(m\right)}=M_{ext}{\delta \left(w_{,2}\right)|}_{x_{2}=0}+{\left(kw+F_{0}\right)\delta w|}_{x_{2}=L_{b}}$$

The first term of (14), that is, the variation of kinetic energy, can be derived as (see Appendix A)
(16)$$\int_{t_{i}}^{t_{f}}\left[\delta T\right]dt=\rho \int_{t_{i}}^{t_{f}}\left\{\int_{\partial \tau }{-\left[\left(x_{1}^{2}{\ddot{w}}_{,2}\right)\delta w\right]|}_{x_{2}=0}^{x_{2}=L_{b}}ds+\int_{\tau }\left(-\ddot{w}+x_{1}^{2}{\ddot{w}}_{,22}\right)\delta wd\tau \right\}dt$$
where τ and ∂τ are the volume and boundary surface of the prong. The strain energy per unit volume is
(17)$$\frac{1}{2}c_{pq}^{E}S_{p}S_{q}=-\frac{1}{2}{\left[\begin{array}{l} 0\\ S_{2}\\ 0\\ 0\\ 0\\ 0 \end{array}\right]}^{T}\left[\begin{array}{cccccc} c_{11} & c_{12} & c_{13} & c_{14} & 0 & 0\\ c_{12} & c_{22} & c_{23} & c_{24} & 0 & 0\\ c_{13} & c_{23} & c_{33} & c_{34} & 0 & 0\\ c_{14} & c_{24} & c_{34} & c_{44} & 0 & 0\\ 0 & 0 & 0 & 0 & c_{55} & c_{56}\\ 0 & 0 & 0 & 0 & c_{56} & c_{66} \end{array}\right]\left\{\begin{array}{l} 0\\ S_{2}\\ 0\\ 0\\ 0\\ 0 \end{array}\right\}=\frac{1}{2}c_{22}S_{2}{}^{2}$$

Substituting Equation (12) into the second term of Hamilton’s principle (14) gives
(18)$$\begin{array}{l} \int_{t_{i}}^{t_{f}}\int_{\tau }-\delta Ud\tau dt=\int_{t_{i}}^{t_{f}}\left\{\int_{\tau }\delta \left[-\frac{1}{2}c_{pq}^{E}S_{p}S_{q}\right]d\tau \right\}dt=\int_{t_{i}}^{t_{f}}\left\{\int_{\tau }\delta \left[-\frac{1}{2}c_{22}S_{2}{}^{2}\right]d\tau \right\}dt\\ =\int_{t_{i}}^{t_{f}}\left\{\int_{\tau }\delta \left[-\frac{1}{2}c_{22}{\left(-x_{1}w_{,22}\right)}^{2}\right]d\tau \right\}dt=\int_{t_{i}}^{t_{f}}\left\{\int_{\tau }\left[-\left(c_{22}x_{1}{}^{2}w_{,22}\right)\delta w_{,22}\right]d\tau \right\}dt \end{array}$$

The integration by parts of the right-hand side of (18) is given by Equations (A5) and (A6). Finally we have
(19)$$\begin{array}{l} \int_{t_{i}}^{t_{f}}\left\{\int_{\tau }\delta \left[-\frac{1}{2}c_{pq}^{E}S_{p}S_{q}\right]d\tau \right\}dt\\ =\int_{t_{i}}^{t_{f}}c_{22}\left\{{\int_{\partial \tau }\left[-\left(x_{1}{}^{2}w_{,22}\right)\delta w_{,2}+\left(x_{1}{}^{2}w_{,222}\right)\delta w\right]|}_{x_{2}=0}^{x_{2}=L_{b}}dS+\int_{\tau }\left[-\left(x_{1}{}^{2}w_{,2222}\right)\delta w\right]d\tau \right\}dt \end{array}$$

The substitution of Equations (15), (16), and (19) into the Hamilton’s principle (14) results in
(20)$$\begin{array}{l} \int_{t_{i}}^{t_{f}}\{\int_{t_{i}}^{t_{f}}\text{\{}\int_{\tau }\left(eq_{1}\delta w\right)d\tau +\int_{\partial \tau }\left[{\left(B_{1}\delta w+B_{2}\delta w_{,2}\right)|}_{x_{2}=0}^{x_{2}=L_{b}}\right]dS\\ \;+{\left(M_{ext}\delta w_{,2}\right)|}_{x_{2}=0}+{\left(kw+F_{0}\right)\delta w|}_{x_{2}=L_{b}}\}dt=0 \end{array}$$

The variation δw in the volume integration of the first term of Equation (20) is arbitrary and its coefficient $eq_{1}$ must be zero, this leads to the equation of motion of the lower loaded prong as
(21)$$\rho \left(-\ddot{w}+x_{1}{}^{2}{\ddot{w}}_{,22}\right)-c_{22}\left(x_{1}{}^{2}w_{,2222}\right)=0$$

Based on (20), at the left boundary $x_{2}=0$, ${\delta w|}_{x_{2}}=0$ leads to $B_{1}\neq 0$ and ${\delta w_{,2}|}_{x_{2}=0}\neq 0$ leads to ${\left(B_{2}+M_{ext}\right)|}_{x_{2}=0}=0$, which is the left boundary condition
(22a)$$-c_{22}\left(x_{1}{}^{2}w_{,22}\right)+M_{ext}=0,\;\mathrm{at}\;x_{2}=0$$

At the right boundary $x_{2}=L_{b}$, both δw and $\delta w_{,2}$ are not zero, therefore, their coefficients must be zero, which leads to the right boundary conditions as 


(22b)
$$\{\begin{array}{c} -\rho \left(x_{1}{}^{2}{\ddot{w}}_{,2}\right)+c_{22}\left(x_{1}{}^{2}w_{,222}\right)+\left(kw+F_{0}\right)=0,\\ -c_{22}\left(x_{1}{}^{2}w_{,22}\right)=0. \end{array}\;\mathrm{at}\;x_{2}=L_{b}$$


For static equilibrium, all the velocities and accelerations are zero. By setting $\ddot{w}=0$ and ${\ddot{w}}_{,22}=0$, Equations (21) and (22a,b) become the static equation and boundary conditions for static displacement $w^{*}\left(x_{2}\right)$ as
(23)$$c_{22}\left(x_{1}{}^{2}w^{*}{}_{,2222}\right)=0.$$
(24)$$M^{*}{}_{ext}=c_{22}\left(x_{1}{}^{2}w^{*}{}_{,22}\right),\;\mathrm{at}\;x_{2}=0.$$
(25)$$\{\begin{array}{c} c_{22}\left(x_{1}{}^{2}w^{*}{}_{,222}\right)+kw^{*}+F_{0}=0,\\ -c_{22}\left(x_{1}{}^{2}w^{*}{}_{,22}\right)=0. \end{array}\;\mathrm{at}\;x_{2}=L_{b}$$

Substituting (3) into (22a,b) and subtracting (23) from it gives
(26)$$\rho \left(-\ddot{\bar{w}}+x_{1}{}^{2}{\ddot{\bar{w}}}_{,22}\right)-c_{22}\left(x_{1}{}^{2}{\bar{w}}_{,2222}\right)=0$$
and
{(27a)M¯ext=c22(x12w¯,22),at x2=0,(27b)−ρ(x12w¯¨,2)+ c22(x12w¯,222)+kw¯=0,at x2=Lb(27c)−c22(x12w¯,22)=0.at x2=Lb 
where $M_{ext}^{}=M_{ext}^{*}\left(x_{2}\right)+{\bar{M}}_{ext}\left(x_{2},t\right)$. The boundary condition (27c) is independent of the constant force $F_{0}$. It is clear that the natural frequencies of vibration around the static equilibrium state are not affected by the constant $F_{0}$, but influenced by the spring constant k. 

### 2.3. The General Solution of the Lower Loaded Prong

The resonant solution of system (24) is of the form
(28)$$\bar{w}\left(x_{2},t\right)=W\left(x_{2}\right)e^{i\omega t}$$
where ω is the natural frequency and $W\left(x_{2}\right)$ is the mode shape. Substituting (28) into (26) and integrating over the cross-section area in the $x_{1}-x_{3}$ plane yields
(29)$$c_{22}I_{2}W_{,2222}+\rho \omega^{2}I_{2}W_{,22}-\rho \omega^{2}I_{1}W=0$$
where
(30a)$$\begin{array}{c} I_{1}=\int_{-\frac{h}{2}}^{\frac{h}{2}}\int_{-\frac{b_{b}}{2}}^{\frac{b_{b}}{2}}dx_{1}dx_{3}=b_{b}h, \end{array}$$
(30b)$$\begin{array}{c} I_{2}=\int_{-\frac{h}{2}}^{\frac{h}{2}}\int_{-\frac{b_{b}}{2}}^{\frac{b_{b}}{2}}x_{1}{}^{2}dx_{1}dx_{3}=\frac{b_{b}^{3}h}{12}\;. \end{array}$$

The mode shape is assumed to be of the form W(x2)=Reλx2, and substituting this into (29) gives the characteristic equation
(31)$$c_{22}I_{2}\lambda^{4}+\rho \omega^{2}I_{2}\lambda^{2}-\rho \omega^{2}I_{1}=0$$

The solution of (31) is
(32)$$\lambda^{2}=\frac{-\rho \omega^{2}I_{2}\pm \sqrt{{\left(\rho \omega^{2}I_{2}\right)}^{2}+4\left(c_{22}I_{2}\right)\left(\rho \omega^{2}I_{1}\right)}}{2\left(c_{22}I_{2}\right)}$$

The general solution of (29) is
(33)$$W\left(x_{2}\right)=R_{1}\cosh\xi_{1}x_{2}+R_{2}\sinh\xi_{1}x_{2}+R_{3}\cos\xi_{2}x_{2}+R_{4}\sin\xi_{2}x_{2}$$
where $\xi_{1}$ and $\xi_{2}$ are related to the natural frequency ω by
(34a)$$\begin{array}{c} \xi_{1}=\sqrt{\frac{-\rho \omega^{2}I_{2}+\sqrt{\Gamma }}{2\left(c_{22}I_{2}\right)}}, \end{array}$$
(34b)$$\begin{array}{c} \xi_{2}=\;\sqrt{\frac{\rho \omega^{2}I_{2}+\sqrt{\Gamma }}{2\left(c_{22}I_{2}\right)}} \end{array}$$
and
(35)$$\Gamma ={\left(\rho \omega^{2}I_{2}\right)}^{2}+4\left(c_{22}I_{2}\right)\left(\rho \omega^{2}I_{1}\right)$$

## 3. The Deformation and Equations of Motion of the Proof Mass

### 3.1. The Deformation of the Proof Mass 

When the QTF is in the state of free vibration at its in-plane anti-phase natural frequency, the two prongs vibrate in opposite directions with the same magnitude. These two prongs will exert shear force and bending moment of opposite direction and equal magnitude on the proof mass, as shown in Figure 4. When the QTF is put closely above the tested sample, the tip-sample interaction force will exert additional moment and shear to the proof mass at their conjunctive area. The net shear force and moment in the interface is shown in Figure 5, and they satisfy
(36)$$Q_{2}=Q_{1}+F,\;M_{2}=M_{1}+M_{F},$$
where $M_{F}=FL_{b}$, F is the spring force acting on the tip of the lower prong. 

#### 3.1.1. Deformation Due to the Bending Moment of the Tip-Sample Interaction

The deformation of the proof mass subject to the bending moment $M_{F}$ could be of bending type or of pure shearing type. We have adopted the type of pure shear deformation in this paper, as shown in Figure 6. The cross-section is inclined by an angle $\beta \left(x_{2},t\right)$.

The displacement of this pure shearing is
(37)$$u_{1}^{\prime }=0,\;u_{2}^{\prime }=-x_{1}\beta ,\;u_{3}^{\prime }=0.$$

#### 3.1.2. The Wrapping Deformation of the Proof Mass

The cross-section experiences wrapped deformation in the anti-phase mode, as shown in Figure 7. We assume that the rotation angle θ of any point $\left(x_{1},x_{2}\right)$ in the cross-section is linearly proportional to the distance $x_{1}$, that is, $\theta =x_{1}\psi \left(x_{2},t\right)$. We assume that there is no axial displacement at the conjunctive point $x_{1}=d/2$, i.e., $u_{2}^{\prime \prime }\left(d/2,x_{2},t\right)=0.$ Then we have
$$u_{2}^{\prime \prime }\left(x_{1},x_{2},t\right)-u_{2}^{\prime \prime }\left(d/2,x_{2},t\right)=\int_{\frac{d}{2}}^{x_{1}}x_{1}\psi \left(x_{2},t\right)dx_{1}=\left(\frac{1}{2}x_{1}^{2}-\frac{d^{2}}{8}\right).$$

So the displacement field of the wrapping deformation is
(38)$$u_{1}^{\prime \prime }=0,\;u_{2}^{\prime \prime }=\left(\frac{1}{2}x_{1}{}^{2}-\frac{d^{2}}{8}\right)\psi \left(x_{2},t\right),\;u_{3}^{\prime \prime }=0.$$

By rotating the wrapping displacement field around the $x_{3}$ axis by an angle $-\beta \left(x_{2},t\right)$, the components resolved along the $x_{1}-x_{2}-x_{3}$ coordinates are obtained as
(39)$$\left\{\begin{array}{c} u_{1}^{\left(w\right)}\\ \begin{array}{c} u_{2}^{\left(w\right)}\\ u_{3}^{\left(w\right)} \end{array} \end{array}\right\}=\left[\begin{array}{cc} \cos\beta & \begin{array}{cc} \sin\beta & 0 \end{array}\\ \begin{array}{c} -\sin\beta \\ 0 \end{array} & \begin{array}{cc} \begin{array}{c} \cos\beta \\ 0 \end{array} & \begin{array}{c} 0\\ 1 \end{array} \end{array} \end{array}\right]\left\{\begin{array}{c} u_{1}^{\prime \prime }\\ \begin{array}{c} u_{2}^{\prime \prime }\\ u_{3}^{\prime \prime } \end{array} \end{array}\right\}=\left\{\begin{array}{c} \psi \left(x_{2},t\right)\sin\beta \left(x_{2},t\right)\left[\frac{x_{1}^{2}}{2}-\frac{d^{2}}{8}\right]\\ \cos\beta \left(x_{2},t\right)\psi \left(x_{2},t\right)\left[\frac{x_{1}^{2}}{2}-\frac{d^{2}}{8}\right]\\ 0 \end{array}\right\}$$

If we assume a small deformation, thus sinβ≈β and cosβ≈1, and drop the nonlinear term, we get
(40)$$u_{1}^{\left(w\right)}=0,\;u_{2}^{\left(w\right)}=\psi \left(x_{2},t\right)\left(\frac{x_{1}^{2}}{2}-\frac{d^{2}}{8}\right),\;u_{3}^{\left(w\right)}=0.$$

Equation (38) plus equation (40) gives the total displacement of the proof mass
(41)$$u_{1}=0,\;u_{2}=-x_{1}\beta \left(x_{2},t\right)+\left(\frac{x_{1}^{2}}{2}-\frac{d^{2}}{8}\right)\psi \left(x_{2},t\right),\;u_{3}=0$$

The strain field of the proof mass is
(42)$$\begin{array}{ccc} S_{1}=0, & S_{2}=-x_{1}\beta_{,2}+\left(\frac{x_{1}^{2}}{2}-\frac{d^{2}}{8}\right)\psi_{,2}, & S_{3}=0,\\ S_{4}=0, & S_{5}=0, & S_{6}=-\beta +x_{1}\psi . \end{array}$$

The strain energy per unit volume is
(43)$$\frac{1}{2}c_{pq}^{E}S_{p}S_{q}=\frac{1}{2}{\left[\begin{array}{l} 0\\ S_{2}\\ 0\\ 0\\ 0\\ S_{6} \end{array}\right]}^{T}\left[\begin{array}{cccccc} c_{11} & c_{12} & c_{13} & c_{14} & 0 & 0\\ c_{12} & c_{22} & c_{23} & c_{24} & 0 & 0\\ c_{13} & c_{23} & c_{33} & c_{34} & 0 & 0\\ c_{14} & c_{24} & c_{34} & c_{44} & 0 & 0\\ 0 & 0 & 0 & 0 & c_{55} & c_{56}\\ 0 & 0 & 0 & 0 & c_{56} & c_{66} \end{array}\right]\left\{\begin{array}{l} 0\\ S_{2}\\ 0\\ 0\\ 0\\ S_{6} \end{array}\right\}=\frac{1}{2}\left(c_{22}S_{2}{}^{2}+c_{66}S_{6}{}^{2}\right)$$

### 3.2. Equations of Motion of the Proof Mass 

The first term of the Hamilton’s principle (14) with the use of (41) is carried out as
(44)$$\begin{array}{l} \int_{t_{i}}^{t_{f}}\left\{\delta T\right\}dt=\int_{\tau }\rho \left\{-\int_{t_{i}}^{t_{f}}{\ddot{U}}_{i}\delta U_{i}dt\right\}d\tau \\ \;=\int_{\tau }\rho \left\{-\int_{t_{i}}^{t_{f}}\left[{\ddot{U}}_{1}\delta U_{1}+{\ddot{U}}_{2}\delta U_{2}+{\ddot{U}}_{3}\delta U_{3}\right]dt\right\}d\tau \\ \;=\int_{\tau }\rho \{-\int_{t_{i}}^{t_{f}}\left[-x_{1}\ddot{\beta }\left(x_{2},t\right)+\left(\frac{1}{2}x_{1}^{2}-\frac{d^{2}}{8}\right)\ddot{\psi }\left(x_{2},t\right)\right]\\ \;\times \left[-x_{1}\delta \beta \left(x_{2},t\right)+\left(\frac{1}{2}x_{1}^{2}-\frac{d^{2}}{8}\right)\delta \psi \left(x_{2},t\right)\right]dt\}d\tau \end{array}$$

The second term of Hamilton’s principle with the use of (43) is developed out as
(45)$$\begin{array}{l} \int_{t_{i}}^{t_{f}}\int_{\tau }-\delta Ud\tau dt==\int_{t_{i}}^{t_{f}}\left\{\int_{\tau }\delta \left[-\frac{1}{2}c_{pq}^{E}S_{p}S_{q}\right]d\tau \right\}dt=\\ \int_{t_{i}}^{t_{f}}\{\int_{\partial \tau }[\left(-c_{22}{\left(\frac{1}{2}x_{1}^{2}-\frac{d^{2}}{8}\right)}^{2}\psi_{,2}+c_{22}\left(\frac{1}{2}x_{1}^{2}-\frac{d^{2}}{8}\right)x_{1}\beta_{,2}\right)\delta \psi \\ {+\left(c_{22}\left(\frac{1}{2}x_{1}^{2}-\frac{d^{2}}{8}\right)x_{1}\psi_{,2}-c_{22}x_{1}{}^{2}\beta_{,2}\right)\delta \beta ]}_{x_{2}=0}^{x_{2}=L_{m}}dS \end{array}\;\begin{array}{l} \;+\int_{\tau }[\left(c_{22}{\left(\frac{1}{2}x_{1}^{2}-\frac{d^{2}}{8}\right)}^{2}\psi_{,22}-c_{22}\left(\frac{1}{2}x_{1}^{2}-\frac{d^{2}}{8}\right)x_{1}\beta_{,22}-c_{66}\left(x_{1}\psi -\beta \right)x_{1}\right)\delta \psi \\ +\left(-\left(\frac{1}{2}x_{1}^{2}-\frac{d^{2}}{8}\right)c_{22}x_{1}\psi_{,22}+c_{22}x_{1}{}^{2}\beta_{,22}+c_{66}\left(x_{1}\psi -\beta \right)\right)\delta \beta ]d\tau \}dt \end{array}$$

The virtual work done on the proof mass has two parts. One comes from the product of the moment $M_{1}$ as shown in Figure 4 and the virtual rotation $\delta \theta =x_{1}\delta \psi$ of the cross-section as shown in Figure 7 at $x_{1}=\pm d/2$. The other comes from the product of the moment $FL_{b}$ caused by the interaction force and the virtual rotation δβ of the cross-section due to the pure shearing deformation, that is,
(46)$$\begin{array}{l} \delta W^{\left(m\right)}=\delta W_{\mathrm{int}erface}^{\left(m\right)}=\delta W_{warping}^{\left(m\right)}+\delta W_{\mathrm{Inclination}}^{\left(m\right)}\\ =\left(\left(-M_{1}\right)\delta \left(-\frac{d}{2}\psi \right)|{}_{x_{2}=L_{m},,x_{1}=-\frac{d}{2}}+M_{1}\delta \left(\frac{d}{2}\psi \right)|{}_{x_{2}=L_{m},x_{1}=\frac{d}{2}}\right)+FL_{b}\delta \beta |{}_{x_{2}=L_{m}}\\ =\left[2M_{1}\delta \left(\frac{d}{2}\psi \right)+FL_{b}\delta \beta \right]|{}_{x_{2}=L_{m}} \end{array}$$
where $L_{m}$ and $b_{m}$ are the length and width of the proof mass, respectively. Substituting Equations (44)–(46) into the Hamilton’s principle (14) yields
(47)$$\begin{array}{l} \int_{t_{i}}^{t_{f}}\{\int_{\tau }\left(\bar{e}{\bar{q}}_{1}\cdot \delta \psi +\bar{e}{\bar{q}}_{2}\cdot \delta \beta \right)d\tau \\ +\int_{\partial \tau }\left[{\left({\bar{B}}_{1}\cdot \delta \psi +{\bar{B}}_{2}\cdot \delta \beta \right)|}_{x_{2}=0}^{x_{2}=L_{m}}\right]dS+\left[M_{1}d\delta \psi +FL_{b}\delta \beta \right]|{}_{x_{2}=L_{m}}\}dt=0 \end{array}$$

In the volume integration of (47) the variations δψ and δβ are arbitrary, that their coefficients $\bar{eq_{1}}$ and $\bar{eq_{2}}$ must be zero gives the two equations of motion as
(48)$$\begin{array}{l} c_{22}{\left(\frac{1}{2}x_{1}^{2}-\frac{d^{2}}{8}\right)}^{2}\psi_{,22}-c_{22}\left(\frac{1}{2}x_{1}^{2}-\frac{d^{2}}{8}\right)x_{1}\beta_{,22}-c_{66}\left(x_{1}\psi -\beta \right)x_{1}\\ +\rho x_{1}\left(\frac{1}{2}x_{1}{}^{2}-\frac{d^{2}}{8}\right)\ddot{\beta }\left(x_{2},t\right)-\rho {\left(\frac{1}{2}x_{1}{}^{2}-\frac{d^{2}}{8}\right)}^{2}\ddot{\psi }\left(x_{2},t\right)=0, \end{array}$$
(49)$$\begin{array}{l} -\left(\frac{1}{2}x_{1}^{2}-\frac{d^{2}}{8}\right)c_{22}x_{1}\psi_{,22}+c_{22}x_{1}{}^{2}\beta_{,22}+c_{66}\left(x_{1}\psi -\beta \right)\\ -\rho x_{1}{}^{2}\ddot{\beta }\left(x_{2},t\right)+\rho x_{1}\left(\frac{1}{2}x_{1}{}^{2}-\frac{d^{2}}{8}\right)\ddot{\psi }\left(x_{2},t\right)=0 \end{array}$$

The left boundary conditions of the proof mass are
(50)$$\psi =0,\beta =0,\;\mathrm{at}\;x_{2}=0.$$

The natural boundary at the right end of the proof mass can be derived from (47). 

The variations δψ and δβ at $x_{2}=L_{m}$ are arbitrary, so their coefficients $\left({\bar{B}}_{1}+M_{1}d\right)$ and $\left({\bar{B}}_{2}+FL_{b}\right)$ must be zero, this leads to right boundary conditions
{(51a)−c22(12x12−d28)2ψ,2+c22(12x12−d28)x1β,2+M1d=0,(51b)c22(12x12−d28)x1ψ,2−c22x12β,2+FLb=0.

The integration of Equation (51a,b) over the cross-section area in $x_{1}-x_{3}^{}$ plane gives
(52)$$\{\begin{array}{c} -c_{22}{\bar{I}}_{1}\psi_{,2}+{\bar{I}}_{5}M_{1}d=0,\\ -c_{22}{\bar{I}}_{3}\beta_{,2}+{\bar{I}}_{5}FL_{b}=0. \end{array}\mathrm{at}\;x_{2}=L_{m}$$

### 3.3. General Solution of the Proof Mass

Integrating the Equations (48) and (49) of motion over the cross-section area gives
(53)$$\{\begin{array}{c} c_{22}{\bar{I}}_{1}\psi_{,22}-c_{66}{\bar{I}}_{3}\psi -\rho {\bar{I}}_{1}\ddot{\psi }=0,\\ c_{22}{\bar{I}}_{3}\beta_{,22}-c_{66}{\bar{I}}_{5}\beta -\rho {\bar{I}}_{3}\ddot{\beta }=0, \end{array}$$
where
(54a)$${\bar{I}}_{1}=\int_{-\frac{h}{2}}^{\frac{h}{2}}\int_{-\frac{b_{m}}{2}}^{\frac{b_{m}}{2}}{\left(\frac{1}{2}x_{1}^{2}-\frac{d^{2}}{8}\right)}^{2}dx_{1}dx_{3}=\frac{h}{960}\left(3b_{m}{}^{5}-10b_{m}{}^{3}d^{2}+15b_{m}d^{4}\right),$$
(54b)$$\begin{array}{c} {\bar{I}}_{3}=\int_{-\frac{h}{2}}^{\frac{h}{2}}\int_{-\frac{b_{m}}{2}}^{\frac{b_{m}}{2}}x_{1}^{2}dx_{1}dx_{3}=\frac{b_{m}^{3}h}{12}, \end{array}$$
(54c)$$\begin{array}{c} \;{\bar{I}}_{5}=\int_{-\frac{h}{2}}^{\frac{h}{2}}\int_{-\frac{b_{m}}{2}}^{\frac{b_{m}}{2}}dx_{1}dx_{3}=b_{m}h. \end{array}$$

Equations (52) and (53) reveal that variables ψ and β are uncoupled, due to the fact that the nonlinear coupling term in the displacement field (39) is neglected for small amplitude vibration. The resonant solution is of the form
(55a)$$\begin{array}{c} \psi \left(x_{2},t\right)=\Psi \left(x_{2}\right)e^{i\omega t}, \end{array}$$
(55b)$$\begin{array}{l} \beta \left(x_{2},t\right)=\Theta \left(x_{2}\right)e^{i\omega t}. \end{array}$$

Substituting (55) into (53) gives
{(56a)c22I¯1Ψ,22−c66I¯3Ψ+ρω2I¯1Ψ=0,(56b)c22I¯3Θ,22−c66I¯5Θ+ρω2I¯3Θ=0 .

The solution of the mode shape for wrapping vibration is of the form $\Psi \left(x_{2}\right)=\bar{R}e^{\lambda x_{2}}$. Substituting this into (56a) gives
(57)$$c_{22}{\bar{I}}_{1}\lambda_{1}{}^{2}-\left(c_{66}{\bar{I}}_{3}-\rho \omega^{2}{\bar{I}}_{1}\right)=0$$

The solution is $\lambda_{1}{}^{2}=\left(c_{66}{\bar{I}}_{3}-\rho \omega^{2}{\bar{I}}_{1}\right)/c_{22}{\bar{I}}_{1}.$ The mode shape is
(58)$$\Psi \left(x_{2}\right)={\bar{R}}_{1}\cosh\xi_{3}x_{2}+{\bar{R}}_{2}\sinh\xi_{3}x_{2}$$
where
(59)$$\xi_{3}=\sqrt{\frac{-\left(c_{66}{\bar{I}}_{3}-\rho \omega^{2}{\bar{I}}_{1}\right)}{c_{22}{\bar{I}}_{1}}}$$

Similarly, the mode shape of Equation (56b) is
(60)$$\Theta \left(x_{2}\right)={\bar{R}}_{3}\cosh\xi_{4}x_{2}+{\bar{R}}_{4}\sinh\xi_{4}x_{2},$$
where $\xi_{4}$ is related to the natural frequency ω by
(61)$$\xi_{4}=\sqrt{\frac{-\left(c_{66}{\bar{I}}_{5}-\rho \omega^{2}{\bar{I}}_{3}\right)}{c_{22}{\bar{I}}_{3}}}$$

## 4. Anti-Phase Resonant Frequency of QTF 

In this section, the three components of the QTF, one free prong, one loaded prong, and the proof mass, are considered as a whole. Their individual general solutions were obtained in previous sections, with the twelve coefficients of mode shapes and the resonant frequency left unknown, to be determined. When the twelve boundary and interface conditions are imposed, the anti-phase resonant frequency can be evaluated. Also, the frequency shift corresponding to the spring constant k (or the negative force gradient) can be found. Three different sets of coordinates need to be used as shown in Figure 8. 

### 4.1. Mode Shapes with Undetermined Coefficients 

The mode shape of the upper free prong is almost similar to Equation (33)
(62)$$W^{\left(2\right)}\left(x_{2}^{\left(2\right)}\right)=R_{1}{}^{\left(2\right)}\cosh\xi_{1}x_{2}^{\left(2\right)}+R_{2}{}^{\left(2\right)}\sinh\xi_{1}x_{2}^{\left(2\right)}+R_{3}{}^{\left(2\right)}\cos\xi_{2}x_{2}^{\left(2\right)}+R_{4}{}^{\left(2\right)}\sin\xi_{2}x_{2}^{\left(2\right)}$$
where $\xi_{1}$ and $\xi_{2}$ are functions of resonant frequency implicitly as
(63)$$\xi_{1}=\sqrt{\frac{-\rho \omega^{2}I_{2}+\sqrt{\Gamma }}{2\left(c_{22}I_{2}\right)}},\;\xi_{2}=\pm \sqrt{\frac{\rho \omega^{2}I_{2}+\sqrt{\Gamma }}{2\left(c_{22}I_{2}\right)}}$$
where Γ is given by (35), coefficients $\left(R_{1}^{\left(2\right)},R_{2}^{\left(2\right)},R_{3}^{\left(2\right)},R_{4}^{\left(2\right)}\right)$ are to be determined. 

The boundary conditions at the right free end are the same as Equation (27b,c), but with the interaction force $k\bar{w}$ and external moment $M_{ext}$ removed.
(64a)$$M_{ext}=c_{22}{\left(x_{1}^{\left(2\right)}\right)}^{2}w_{,22}^{\left(2\right)},\;\mathrm{at}\;x_{2}^{\left(2\right)}=0$$
{(64b)−ρ[(x1(2))2w¨,2(2)]+c22[(x1(2))2w,222(2)]=0, at x2(2)=Lb(64c)−c22[(x1(2))2w,22(2)]=0, at x2(2)=Lb

Equation (64b) is the shear-force free condition and (64c) is the moment free condition. 

Integrating (64b,c) over the cross-section area and using the mode shape (62) gives
(65)$$\{\begin{array}{c} \rho \omega^{2}I_{2}W^{\left(2\right)}{}_{,2}+c_{22}I_{2}W^{\left(2\right)}{}_{,222}=0,\\ -c_{22}I_{2}W^{\left(2\right)}{}_{,22}=0. \end{array}\;\mathrm{at}\;x_{2}^{\left(2\right)}=L_{b}$$

The mode shapes of the proof mass for cross-section wrapping and shearing in terms of the newly-defined coordinates are
(66)$$\{\begin{array}{c} \Psi^{\left(1\right)}\left(x_{2}^{\left(1\right)}\right)={\bar{R}}_{1}{}^{\left(1\right)}\cosh\xi_{3}x_{2}^{\left(1\right)}+{\bar{R}}_{2}{}^{\left(1\right)}\sinh\xi_{3}x_{2}^{\left(1\right)},\\ \Theta^{\left(1\right)}\left(x_{2}^{\left(1\right)}\right)={\bar{R}}_{3}{}^{\left(1\right)}\cosh\xi_{4}x_{2}^{\left(1\right)}+{\bar{R}}_{4}{}^{\left(1\right)}\sinh\xi_{4}x_{2}^{\left(1\right)}. \end{array}$$
where $\xi_{3}$ and $\xi_{4}$, being functions of resonant frequency, are given by (59) and (61).

The boundary and interface conditions are given by (50) and (52).

The mode shape of the lower loaded prong in terms of the new coordinates is
(67)$${\bar{W}}^{\left(3\right)}(x_{2}^{\left(3\right)})=R_{1}{}^{\left(3\right)}\cosh\xi_{1}x_{2}^{\left(3\right)}+R_{2}{}^{\left(3\right)}\sinh\xi_{1}x_{2}^{\left(3\right)}+R_{3}{}^{\left(3\right)}\cos\xi_{2}x_{2}^{\left(3\right)}+R_{4}{}^{\left(3\right)}\sin\xi_{2}x_{2}^{\left(3\right)}$$
where the frequency dependent parameters $\xi_{1}$ and $\xi_{2}$ are given by Equation (34a,b). The boundary conditions are
(68a)$${\bar{M}}_{ext}=c_{22}\left(x_{1}^{\left(3\right)}\right){\bar{w}}_{,22}^{\left(3\right)},\;\mathrm{at}\;x_{2}^{\left(3\right)}=0.$$

Integrating Equations (27b,c) over the cross-section area gives
(68b)$$\{\begin{array}{c} \rho \omega^{2}I_{2}{\bar{W}}_{,2}^{\left(3\right)}+c_{22}I_{2}W_{,222}^{\left(3\right)}+kb_{b}h{\bar{W}}^{\left(3\right)}=0,\\ -c_{22}I_{2}W_{,22}^{\left(3\right)}=0. \end{array}\;\mathrm{at}\;x_{2}^{\left(3\right)}=L_{b}$$

### 4.2. The Twelve Boundary and Interface Conditions

The proof mass has no wrapping rotation and shear inclination at the clamped boundary, so
(69a)$$\Psi^{\left(1\right)}\left(x_{2}^{\left(1\right)}=0\right)=0,$$
(69b)$$\Theta^{\left(1\right)}\left(x_{2}^{\left(1\right)}=0\right)=0$$

At the interface, the free prong has no vertical displacement,
(69c)$$W^{\left(2\right)}\left(x_{2}^{\left(2\right)}=0\right)=0.$$

The slope of the proof mass at the interface is equal to that of the free prong, i.e.,
(69d)$${\left(\Theta^{\left(1\right)}-\frac{d}{2}\Psi^{\left(1\right)}\right)|}_{x_{2}^{\left(1\right)}=L_{m},\;x_{1}^{\left(1\right)}=-d/2}=W_{,2}^{\left(2\right)}\left(x_{2}^{\left(2\right)}=0\right)$$

The condition that the moment acting on the proof mass is equal in magnitude to the moment acting on the free prong at the upper interface is not used, because the end surface of the prong is flat while the proof mass is curvilinear. Instead, the condition that the work, $\delta W_{{}_{m2}}^{\left(1\right)}$, done by the moment $M_{1}$ in (51a) on the virtual angular displacement of the proof mass be equal to that, $\delta W_{b}^{\left(2\right)}$, done by the moment $M_{ext}\left(x_{2}^{\left(2\right)}\right)$ in (64a) on the virtual rotation of the prong end face is used here. Note that
$$\delta W_{b}^{\left(2\right)}=\int_{-\frac{h}{2}}^{\frac{h}{2}}\int_{-\frac{b_{b}}{2}}^{\frac{b_{b}}{2}}\left\{\left[-c_{22}\left({\left(x_{1}{}^{\left(2\right)}\right)}^{2}w_{,22}\right)\right]\delta w_{,2}\right\}dx_{1}^{\left(2\right)}dx_{3}^{\left(2\right)}$$
and
$$\begin{array}{l} \delta W_{{}_{m2}}^{\left(1\right)}=\int_{-\frac{h}{2}}^{\frac{h}{2}}\int_{-\frac{d}{2}-\frac{b_{b}}{2}}^{-\frac{d}{2}+\frac{b_{b}}{2}}\{\frac{1}{d}[c_{22}{\left(\frac{1}{2}{\left(x_{1}^{\left(1\right)}\right)}^{2}-\frac{d^{2}}{8}\right)}^{2}\psi_{,2}\\ \;-c_{22}\left(\frac{1}{2}{\left(x_{1}^{\left(1\right)}\right)}^{2}-\frac{d^{2}}{8}\right)x_{1}^{\left(1\right)}\beta_{,2}]\delta \left(-\frac{d}{2}\psi \right)\}dx_{1}^{\left(1\right)}dx_{2}^{\left(1\right)} \end{array}$$

The condition $\delta W_{b}^{\left(2\right)}=\delta W_{{}_{m2}}^{\left(1\right)}$ with the use of $\psi =w_{,2}$ and integration over the contact area leads to
(69e)$$dI_{2}W_{,22}\left(x_{2}{}^{\left(2\right)}=0\right)=-{\bar{I}}_{1}\Psi_{,2}\left(x_{2}{}^{\left(1\right)}=L_{m}\right)$$
where
$I_{2}=\int_{-\frac{h}{2}}^{\frac{h}{2}}\int_{-\frac{b_{b}}{2}}^{\frac{b_{b}}{2}}\left(x_{1}{}^{2}\right)dx_{1}dx_{3}=\frac{1}{12}b_{b}{}^{3}h$,${\bar{I}}_{1}=\int_{-\frac{h}{2}}^{\frac{h}{2}}\int_{-\frac{d}{2}-\frac{b_{b}}{2}}^{-\frac{d}{2}+\frac{b_{b}}{2}}\left[{\left(\frac{1}{2}{\left(x_{1}^{\left(2\right)}\right)}^{2}-\frac{d^{2}}{8}\right)}^{2}\right]dx_{1}^{\left(2\right)}dx_{3}^{\left(2\right)}=\frac{h}{960}\left(3\;b_{b}{}^{5}+20b_{b}{}^{3}d^{2}\right)\;$

The shear force at zero at the free end of the upper prong is
(69f)$$\rho \omega^{2}I_{2}W_{\;,2}^{\left(2\right)}+c_{22}I_{2}W_{,222}^{\left(2\right)}=0,\;\mathrm{at}\;x_{2}^{\left(2\right)}=L_{b}.$$

The bending moment at zero at the free end of the upper prong is
(69g)$$-c_{22}I_{2}W_{\;,22}^{\left(2\right)}=0.\;\;\mathrm{at}\;x_{2}^{\left(2\right)}=L_{b}.$$

The lower prong has no displacement in the $x_{1}^{\left(3\right)}$ direction at the interface
(69h)$$W^{\left(3\right)}\left(x_{2}^{\left(3\right)}=0\right)=0.$$

The slope of the lower prong is equal to that of the proof mass at the lower interface
(69i)$$W_{,2}^{\left(3\right)}\left(x_{2}{}^{\left(3\right)}=0\right)=\Theta +\frac{d}{2}\Psi^{\left(1\right)}\left(x_{2}{}^{\left(1\right)}=L_{m},x_{1}{}^{\left(3\right)}=\frac{d}{2}\right)$$

The virtual work $\delta W_{b}^{\left(3\right)}$ done by the moment in (24) on the virtual rotation of the prong is equal to the work $\delta W_{m3}^{\left(1\right)}$ done by the moment in (51a) on the virtual angular displacement of the proof mass, i.e., the condition $\delta W_{b}^{\left(3\right)}=\delta W_{m3}^{\left(1\right)}$ gives
(69j)$$dI_{2}W_{,22}\left(x_{2}{}^{\left(3\right)}=0\right)={\bar{I}}_{1}\Psi_{,2}\left(x_{2}{}^{\left(1\right)}=L_{m}\right)$$

The balance of the shear force at the right end of the lower loaded prong is
(69k)$$\rho \omega^{2}I_{2}W_{,2}^{\left(3\right)}+c_{22}I_{2}W_{,222}^{\left(3\right)}+kb_{b}hW_{}^{\left(3\right)}=0,\;\mathrm{at}\;x_{2}^{\left(3\right)}=L_{b}$$

The moment-free of the right loaded prong is
(69l)$$-c_{22}I_{2}W_{{}_{,22}}^{\left(3\right)}=0.$$

Substituting the general solutions (62), (66), and (67) into the twelve boundary and interface conditions given by the Equations (69a–l) yields the 12×12 matrix-vector equation
(70)$${\left[M_{ij}\right]}_{12\times 12}\left\{R_{j}\right\}=0$$
where $\left\{R\right\}=\left[\begin{array}{cccc} {\bar{R}}_{1}{}^{\left(1\right)} & \begin{array}{ccc} {\bar{R}}_{2}{}^{\left(1\right)} & {\bar{R}}_{3}{}^{\left(1\right)} & {\bar{R}}_{4}{}^{\left(1\right)} \end{array} & R_{1}{}^{\left(2\right)} & R_{2}{}^{\left(2\right)} \end{array}\begin{array}{cccc} R_{3}{}^{\left(2\right)} & R_{4}{}^{\left(2\right)} & R_{1}{}^{\left(3\right)} & \begin{array}{ccc} R_{2}{}^{\left(3\right)} & R_{3}{}^{\left(3\right)} & R_{3}{}^{\left(3\right)} \end{array} \end{array}\right]$. 

The non-zero elements of the matrix $\left[M_{ij}\right]$ are given in Appendix B. 

For the nontrivial solution of the parametric vector {R} the determinant of the coefficient matrix of Equation (70) must be zero, that is,
(71)$$\det\left[M_{ij}\left(\omega \right)\right]=0$$
this leads to the characteristic equation, which is a nonlinear algebraic equation of natural frequency ω. By solving this characteristic equation numerically, we get the natural frequencies of the in-plane anti-phase resonant modes. 

### 4.3. Results and Discussions

The geometric parameters of QTF are as follows. (1) The length, width, and thickness of the prong are 154 μm, 14 μm, and 7 μm, respectively. (2) The length, width, and thickness of the proof mass are 60 μm, 46 μm, and 7 μm, respectively. The gap between the two prongs is 8 μm. The Hamaker constant for quartz is $8.83\times 10^{-20}\;J$ [25], the radius of the tip is 8 nm, the density of QTF is ρ=2649 kg/mand the elastic stiffness coefficients are given by (10). 

The natural frequency and frequency shift associated with various values of the spring constant are listed in Table 1 through the numerical solution of (71), and are plotted in Figure 9. It is obvious based on Figure 9 that increasing the spring constant will increase the natural frequency, and hence there is a frequency shift, which is approximately a parabolic function of the spring constant. 

Given an initial height z denoted by $z_{j}$, the spring constant $k\left(z_{j}\right)$ and $F_{0}\left(z_{j}\right)$ can be evaluated from Equation (7). The solution of the static equilibrium state given by Equation (25) with the left end clamped is easily obtained as
(72)$$w^{*}\left(z_{j}\right)=\frac{-F_{0}\left(z_{j}\right)}{\left(c_{22}I_{2}-\frac{k\left(z_{j}\right)L_{b}{}^{3}}{3}\right)}(\frac{x_{2}^{3}}{6}-\frac{x_{2}^{2}L_{b}}{2}).$$

During measurement, the vibration amplitude $\bar{w}$ of the prong is controlled as a small constant denoted by $A_{\mathrm{const}}$ (which is called amplitude modulation AFM, or AM-AFM) so as not to touch the sample, the distance $s\left(z_{j}\right)$ between the tip and sample is
(73)$$s\left(z_{j}\right)=z_{j}-w^{*}\left(z_{j}\right)-A_{\mathrm{const}}$$

Then, by solving Equation (71) the natural frequency and frequency shift (Δf) can be obtained for this given initial height $z_{j}$. Varying the initial height means varying the distance $s\left(z_{j}\right)$ and the frequency shift can be plotted as function of $s\left(z_{j}\right)$, as shown in Figure 10 in the far-range regime $20\overset{\circ }{A}\leq s\leq 60\overset{\circ }{A}$. Including the shot-range repulsive regime $\left(3\overset{\circ }{A}\leq s\leq 5\overset{\circ }{A}\right)$, Δf vs. s is plotted in Figure 11. 

As to the experimental work, due to the piezoelectric effect of the single-crystal quartz, QTF can sense the frequency shift. If the tip-sample distance s(z) can be measured, the frequency shift Δf as a function of $s\left(z_{j}\right)$ can be obtained experimentally as
(74)Δf=g(s). 

However, the relationship between the frequency shift and the force gradient is not available from QTF measurement. This hindered us finding the tip-sample force, but it can be obtained by solving the equations of motion established from proper physical modeling. From the equation $s=z-w^{*}\left(z\right)-\bar{w}$ we have a fixed value of z and therefore the fixed value of static deflection $w^{*}\left(z\right)$,
(75)$$\frac{\partial F}{\partial \bar{w}}=\frac{\partial F}{\partial s}\frac{\partial s}{\partial \bar{w}}=-\frac{\partial F}{\partial s}.$$

The solution of (71) gives
(76)$$\Delta f=h\left(\partial F/\partial \bar{w}\right)=h\left(-\partial F/\partial s\right),\;\mathrm{or}\;\frac{\partial F}{\partial s}=-h^{-1}\left(\Delta f\right)$$

With the use of (74) and (75), Equation (76) becomes
(77)$$\frac{\partial F}{\partial s}=-h^{-1}\left(g\left(s\right)\right)=p\left(s\right)$$

The integration of Equation (77) provides the tip-sample force as
(78)$$F\left(s\right)\approx F\left(s\right)-F\left(\infty \right)=\int_{\infty }^{s}p\left(s\right)ds$$

In short, the QTF experiment produces the frequency shift as function of distance s, such as in Equation (74). Physical modeling gives the frequency shift as function of the force gradient, like in Equation (76). Combining these two functions and performing integrations numerically, the tip-sample interaction force as a function of s comes out. Since the interaction force is the negative gradient of the potential, therefore, the integration of Equation (78) again gives a negative potential for the tip-sample interaction. 

### 4.4. Experimental Test for the Natural Frequency

To support the proposed physical model based on elasticity and dynamics, an experimental test for the natural frequency of the first flexural-bending and anti-phase mode was executed. If a QTF of tiny size as the one used in Section 4.3 may not be available at hand, a QTF of moderate size can be obtained from a wristwatch by etching out the epoxy packing. Since the geometrical dimensions influence the natural frequency significantly, the length, width, and gap between the two prongs are measured by microscope as shown in Figure 12, Figure 13 and Figure 14. The length and width of the proof mass are shown in Figure 15 and Figure 16. The thickness of the QTF is measured by using SEM as shown in Figure 17. 

The results of analytical solution, finite element simulation using commercial code COMSOL, and the experimental measurements are listed in Table 2. 

From Table 2, it is clear that the natural frequency predicted by theoretical analysis is closer to the experimental one than that obtained by finite element analysis. The reason for this is that the theoretical result is obtained by solving the equations of motion of the system exactly, so the eigenvalue (natural frequency) and eigenfunction (mode shape) should approach the true solutions. The only error comes from the small discrepancy between our equations of motion and the true equations of motion, while the method of finite element uses the admissible functions to replace the unknown eigenfunctions. The solution process is explained in Appendix C. 

## 5. Conclusions

Although non-contact QTF can provide the frequency shift, due to the tip-sample interaction as a function of their distance, the high resolution images of the tested molecules, atoms, or living cells need precise physical modeling for providing the relationship between the frequency shift and the force gradient. The popular physical model contains a simple harmonic oscillator or two coupled oscillators. However, it encounters the problem of determining the spring constant correctly. This is still controversial among researchers. In the paper, we proposed a new physical model, which models the QTF as elastic bodies instead of point mass oscillator. Based on the elastic theory and analytical dynamics, we can completely solve the problem of finding the relationship between the frequency shift and force gradient. 

## Figures and Tables

**Figure 1 sensors-19-01948-f001:**
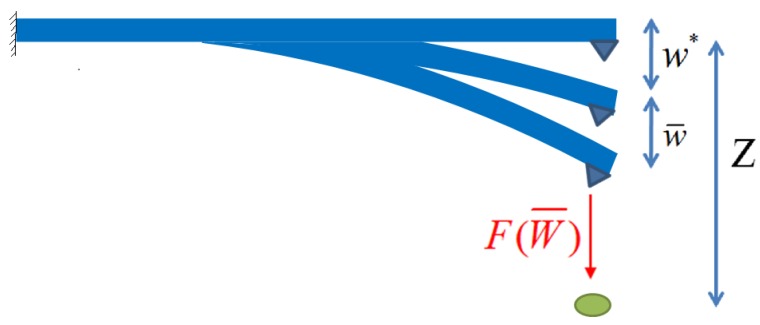
The displacements of the tip in static equilibrium state and vibration state.

**Figure 2 sensors-19-01948-f002:**
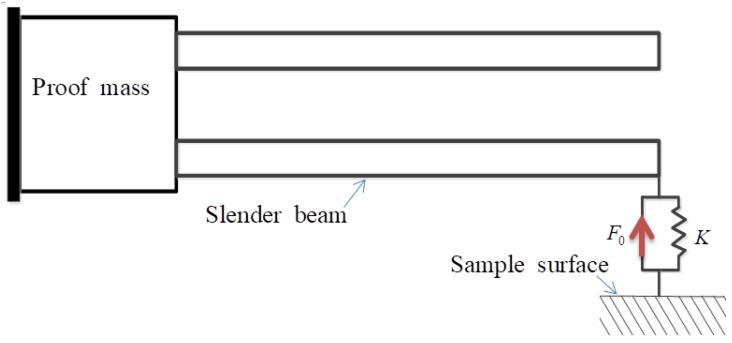
The linearized Lennard-Jones force is equivalent to a spring force $k\bar{w}$ plus a constant force $F_{0}$.

**Figure 3 sensors-19-01948-f003:**
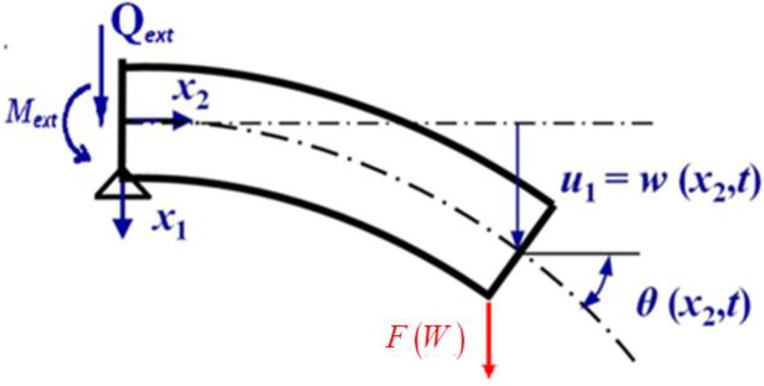
The displacement and loading condition of the lower prong.

**Figure 4 sensors-19-01948-f004:**
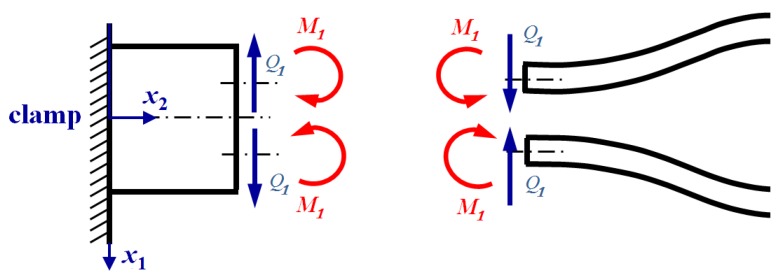
Shear force and bending moment acting on the proof mass and the prongs at the interfacial area in the case of free vibration.

**Figure 5 sensors-19-01948-f005:**
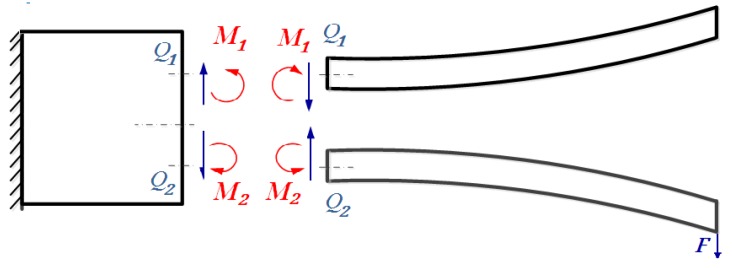
The shear forces and moments acting on the proof mass are exerted by the two prongs and the applied force F.

**Figure 6 sensors-19-01948-f006:**
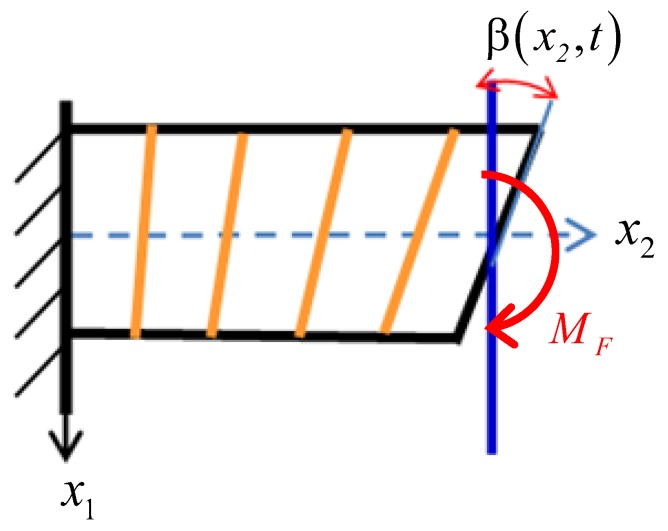
Shear deformation of the proof mass caused by the bending moment $M_{F}$.

**Figure 7 sensors-19-01948-f007:**
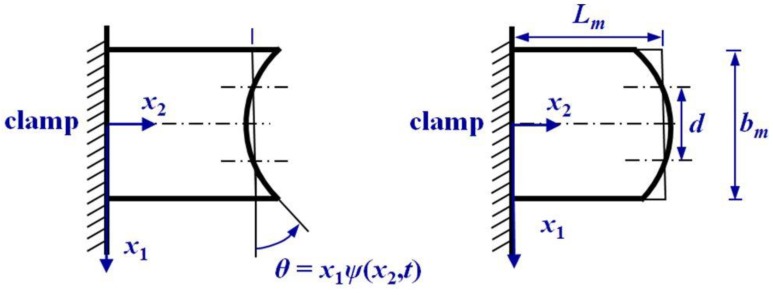
Wrapped curve of the cross-section of the proof mass.

**Figure 8 sensors-19-01948-f008:**
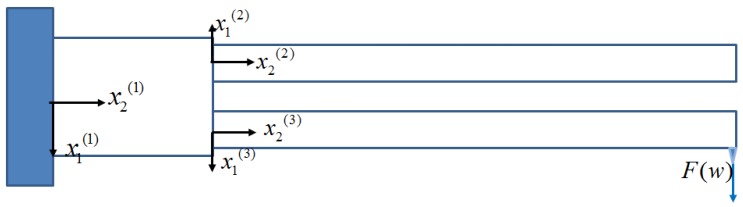
Three sets of coordinates of the cantilever quartz tuning fork (QTF).

**Figure 9 sensors-19-01948-f009:**
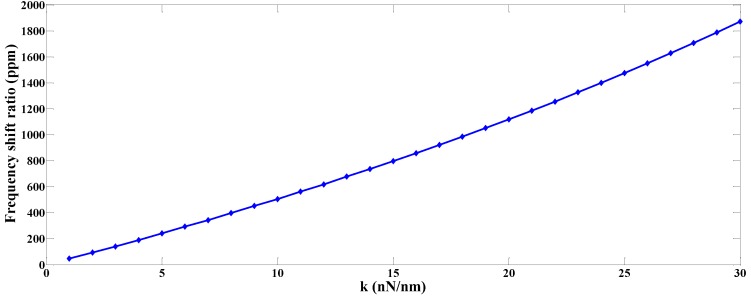
The relationship between the frequency shift and the spring constant.

**Figure 10 sensors-19-01948-f010:**
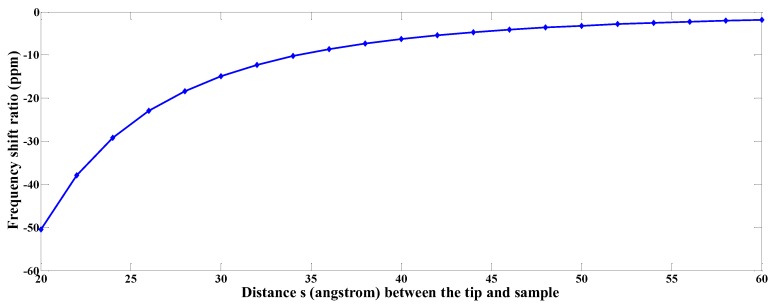
The plot of frequency shift ratio versus the tip-sample distance in far-range regime.

**Figure 11 sensors-19-01948-f011:**
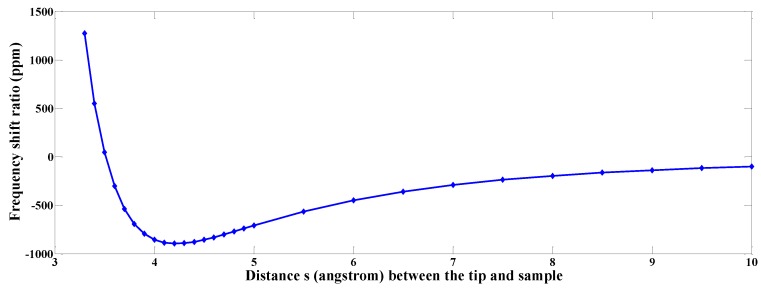
Frequency shift ratio versus the tip-sample distance, including the short-range repulsive regime.

**Figure 12 sensors-19-01948-f012:**
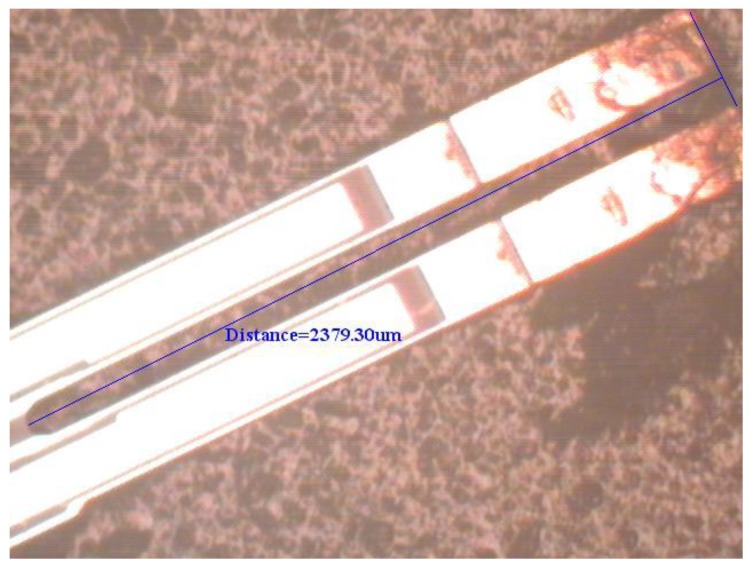
Determination of the length of the prong from a microscope image.

**Figure 13 sensors-19-01948-f013:**
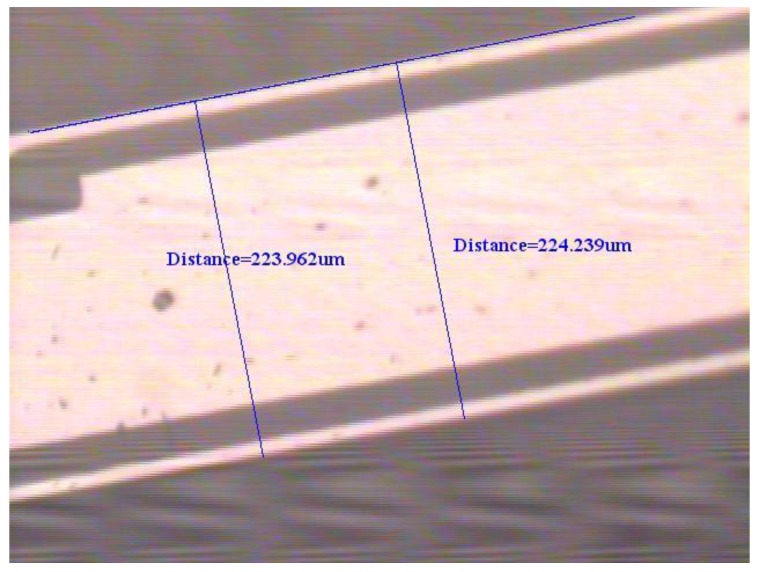
Determination of the width of the prong from a microscope image.

**Figure 14 sensors-19-01948-f014:**
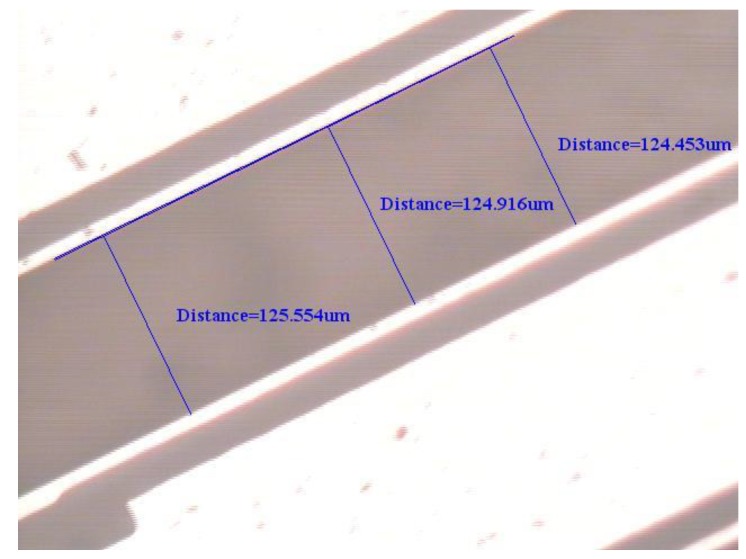
Determination of the gap between the two prongs from a microscope image.

**Figure 15 sensors-19-01948-f015:**
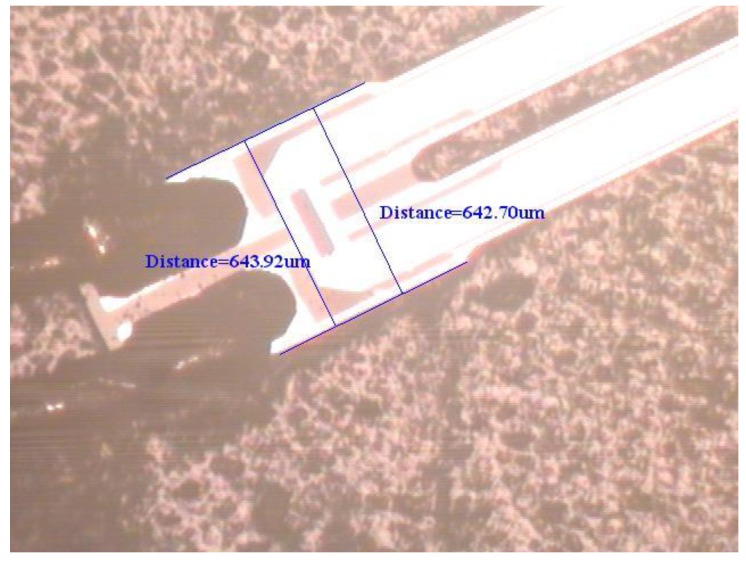
Determination of the width of the proof mass from a microscope image.

**Figure 16 sensors-19-01948-f016:**
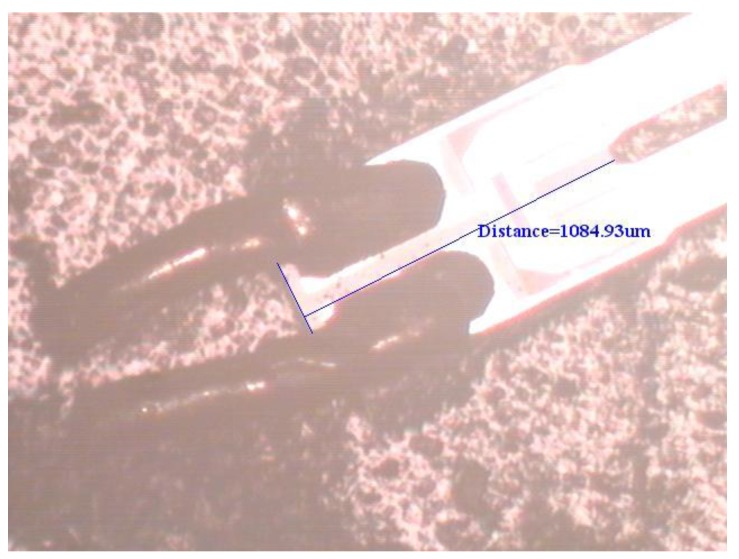
Determination of the length of the proof mass from a microscope image.

**Figure 17 sensors-19-01948-f017:**
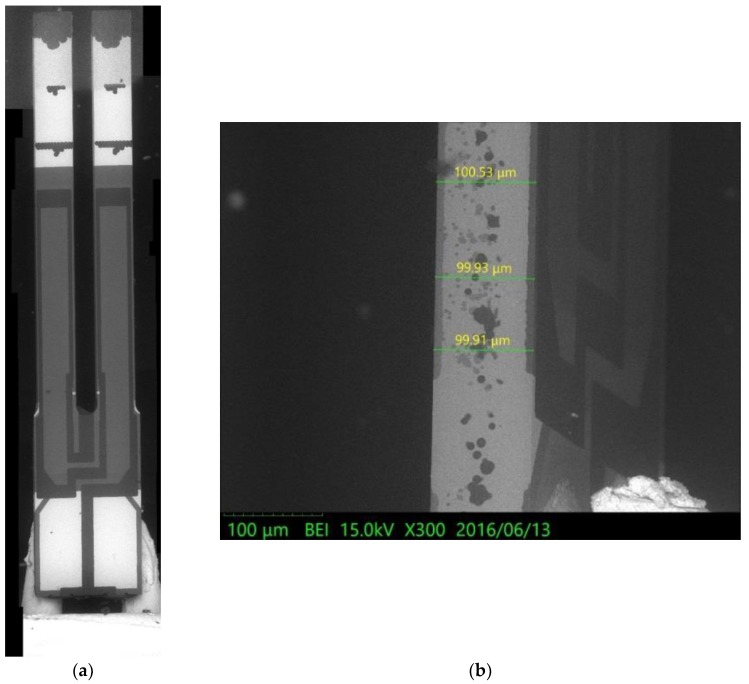
Thickness measurements from an SEM image. (**a**) The top view of the QTF. (**b**). The side view of the QTF.

**Table 1 sensors-19-01948-t001:** Spring constants and natural frequencies of the first anti-phase mode and frequency shift.

$k=\frac{nN}{nm}$	Frequency (kHz)	Frequency Shift (Hz)	Frequency Shift Ratio (ppm)
1	451.219	21	46.5
2	451.24	42	93.1
3	451.262	64	141.8
4	451.284	86	190.6
5	451.307	109	241.5
6	451.33	132	292.5
7	451.353	155	343.4
8	451.377	179	396.6
9	451.402	204	451.9
10	451.426	228	505.1
11	451.452	254	562.6
12	451.477	279	618.0
13	451.504	306	677.7
14	451.53	332	735.3
15	451.558	360	797.2
16	451.586	388	859.2
17	451.614	416	921.1
18	451.643	445	985.3
19	451.673	475	1051.6
20	451.703	505	1118.0
21	451.734	536	1186.5
22	451.765	567	1255.1
23	451.797	599	1325.8
24	451.83	632	1398.8
25	451.864	666	1473.9
26	451.898	700	1549.0
27	451.934	736	1628.6
28	451.969	771	1705.9
29	452.006	808	1787.6
30	452.044	846	1871.5

**Table 2 sensors-19-01948-t002:** Natural frequency of the first in-plane anti-phase mode from experiment, FEM (finite element method), and theoretical analysis.

	Experiment	Analysis	FEM
Frequency (kHz)	32.768 (kHz)	33.367 (kHz)	31.466 (kHz)
Error (%)	-	1.83%	−3.97%

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
