# Peer review of "Analysis of the Frequency Shift versus Force Gradient of a Dynamic AFM Quartz Tuning Fork Subject to Lennard-Jones Potential Force"

_sensors, 2019, doi:10.3390/s19081948_

Round 1

Reviewer 1 Report

The authors describe a method in which they obtain the mechanical properties of a quartz tuning fork (QTF) via an analytical approach based on continuum theory with shear forces and bending moments. The authors moreover introduce an additional external potential which acts on one of the prongs of the QTF. Their goal is to directly relate measured frequency shifts to actual force gradients. The approach is interesting as it is an alternative to computationally more demanding finite-element simulations of the QTF. On the other hand, the harmonic approximation criticized by the authors is sufficient and typically used if the stiffness of the QTF is known. In the approach used by the authors this vale is implicitly extracted from the material’s properties and dimensions of the QTF. I think the authors should address two points prior to publication to improve the value of the article for the reader:

(1)    They should add references to finite element approaches to QTF simulations which can yield similar results, also they should add references to articles which experimentally measure the stiffness of QTFs with, e.g.,
A simple method for the determination of qPlus sensor spring constants” Melcher et al. (2015)
Calibration of quartz tuning fork spring constants for non-contact atomic force microscopy: direct mechanical measurements and simulations” Falter et al. (2014)

(2)    They should assess the quality of their results with respect to experiment: Since the calculations are apparently done with a specific QTF design in mind, it would be easy to compare the actual (experimental) resonance frequency of this type of QTF with the resonance frequency they obtain from their analytical equations. This comparison could emphasize the validity and value of the presented approach.

Author Response

Reply to the first comment:

 We indeed include the two references as required in the manuscript which are numbered  as18 and 19 and have given description from Line 61 to line 68.

Reply to the second comment:

  We have provided the experimentl test for measuring the natural frequency of the first flexural-bending anti-phase mode, which is inserted from line 350 to 402. .

 We also compare the result of theoretical one and that of the finite element anlaysis to the ezperimental one.

 Our result is closer to tne experimental one than that of FEA.

Reviewer 2 Report

This manuscript presents an analytical study of the relationship between the frequency-shift and force gradient of self-sensing and self-actuating quartz tuning fork (QTF), through the use of Hamilton’s principle to obtain the equations of motion of the QTF that is subject to Lennard-Jones potential force.  While the results are of practical utility, the authors may want to validate the model with experimental results.  The manuscript would also benefit from professional editing (e.g., "mas" -> "mass" in the abstract, inconsistent font size in the equations, grammatical issues, and etc).

Author Response

Reply to the first comment:

 We have provided the content of experiemtnal test for measuring the natural freqnency of the QTF and give the comparision of the result of our theoretical analyisis and that from finite element analysis to the experimental one, which are given from line 350 to 402 in the manuscript.

Reply to the second comment:

  We have revised the English and correct the error in the numbering of equations, which are all  marked in yellow background.
